

# Five genes influenced by obesity may contribute to the development of thyroid cancer through the regulation of insulin levels

Jiaming Chen[1], Hongbao Cao[2,3], Meng Lian[1] and Jugao Fang[1]

[1] Department of Otorhinolaryngology Head and Neck Surgery, Beijing Tongren Hospital, Capital Medical University, Beijing, China
[2] Department of Psychiatry, First Hospital/First Clinical Medical College of Shanxi Medical University, Taiyuan, China
[3] School of Systems Biology, George Mason University, Fairfax, VA, United States of America

## ABSTRACT

Previous studies indicate that obesity is an important contributor to the proceeding of thyroid cancer (TC) with limited knowledge of the underlying mechanism. Here, we hypothesize that molecules affected by obesity may play roles in the development of TC. To test the hypothesis above, we first conducted a large-scale literature-based data mining to identify genes influenced by obesity and genes related to TC. Then, a mega-analysis was conducted to study the expression changes of the obesity-specific genes in the case of TC, using 16 independent TC array-expression datasets (783 TC cases and 439 healthy controls). After that, pathway analysis was performed to explore the functional profile of the selected target genes and their potential connections with TC. We identified 1,036 genes associated with TC and 534 regulated by obesity, demonstrating a significant overlap ($N = 176$, $p\text{-value} = 4.07e{-}112$). Five out of the 358 obesity-specific genes, FABP4, CFD, GHR, TNFRSF11B, and LTF, presented significantly decreased expression in TC patients (LFC$<{-}1.44$; and $p\text{-value} < 1e^{-7}$). Multiple literature-based pathways were identified where obesity could promote the pathologic development of TC through the regulation of these five genes and INS levels. The five obesity genes uncovered could be novel genes that play roles in the etiology of TC through the modulation of INS levels.

## INTRODUCTION

Thyroid cancer (TC) develops from the tissues of the thyroid gland and becomes the fastest-growing cancer of all malignancies (*Wolin, Carson & Colditz, 2010*). Approximately 20% of all types of cancers might be caused by excessive weight (overweight or obesity) (*Wolin, Carson & Colditz, 2010*). Epidemiologic research suggested that there could be a positive correlation between the increased incidence of both obesity and TC in the past decades (*Ogden et al., 2007*). The hypothesis has been supported by multiple studies with different methodologies, including cohort study, pooled analysis, and meta-analysis

Corresponding author
Jugao Fang, jugaofang@gousinfo.com

(*Engeland et al., 2006*; *Kitahara et al., 2011*; *Zhao et al., 2012*). However, inconsistent results were represented, which could be due to the unbalanced sex in the TC sample (*Meinhold et al., 2010*), different sample population regions (*Oh, Yoon & Shin, 2005*), and the lack of adjustment of other influential factors (*Meinhold et al., 2010*; *Engeland et al., 2006*).

Nevertheless, many studies have been made to explore the mechanisms underlying the obesity-TC relationship, taking the advances made by molecular biologists (*Nannipieri et al., 2009*; *Hard, 1998*; *Liu et al., 2012*; *Ozgen et al., 2009*; *Stassi et al., 2003*; *Iyengar et al., 2017*; *Park et al., 2016*). Some studies showed evidence that regional obesity and a tendency to weight gain were associated with the variations in thyroid function. For example, an increase of triiodothyronine ($T_3$) levels was observed in obese subjects (*Nannipieri et al., 2009*). TSH is the major stimulator of thyrocyte proliferation; the high level of this hormone could be directly involved in thyroid carcinogenesis in obese subjects (*Hard, 1998*). In addition, multiple genetic and epigenetic alterations of obesity have been reported as pathophysiological important, with many of them also identified as genetic targets for early diagnosis, prognosis or the therapeutic response to the treatment of TC (*Liu et al., 2012*; *Ozgen et al., 2009*; *Stassi et al., 2003*; *Iyengar et al., 2017*; *Park et al., 2016*). For instance, separate sets of studies analyzing the pro- and anti-inflammatory cytokines, TNF-$\alpha$, IL-6, and IL-10, which are part of the obesity-associated secretory phenotype, showed their roles in the deterioration or treatment of TC (*Liu et al., 2012*; *Ozgen et al., 2009*; *Stassi et al., 2003*; *Iyengar et al., 2017*). However, the mechanism regarding this obesity-promoting-TC relationship remains mostly unclear.

Taken together, these observations indicated the presence of some not-yet discovered connections between obesity and TC. In this study, we attempted to use a system biology approach to identify the not-yet discovered connections between both diseases, including the data mining of disease-gene relation data, the analysis of molecular pathways, and a mega-analysis of existing expression datasets. The integrated analysis of multiple modalities of data has been proven to be an effective way for disease mechanism study (*Liu et al., 2019*; *Zhang et al., 2019*; *Lian et al., 2019*).

## METHODS AND MATERIALS

This study was organized as follows. First, the large-scale literature-based TC-gene and obesity-gene relations data were mined, through which obesity- and TC-genes were identified and compared. Then, a mega-analysis was conducted to test genes that were regulated by obesity but not implicated with TC. After that, a literature-based pathway analysis and a gene set enrichment analysis (GSEA) were performed to identify the potential functional network connecting the selected molecules and TC and the biological profile of these molecules.

### Literature-based relation data

Literature-based genetic relation data was conducted by using Pathway Studio (http://www.pathwaystudio.com), and results were organized into Supplemental Information 1. Besides the full lists of genes, we also presented the information of supporting references for each disease-gene relation, including titles of the references and the related sentences where

the disease-gene relationships were identified (Obesity_TC → TC genes and Obesity_TC → Obesity_genes). The information could be used to locate a detailed description of how a candidate gene is associated with obesity and/or TC. To increase the reliability of the obesity affected genes, we selected the obesity-gene relationships with at least three supporting references and with a specific polarity (positive or negative regulation).

## Selection of TC-RNA expression datasets

We search all TC array-expression datasets available at GEO. After the initial search with keyword 'thyroid cancer', we identified 91 expression datasets for TC. Then the following criteria were applied to fulfill the purpose of this study, including (1) The data organism is Homo sapiens; (2) The data type is RNA expression; (3) The sample size is no less than 10, and (4) the studies are limited to TC cases vs. healthy controls design.

## Mega-analysis and target selection

A mega-analysis was conducted for each of these genes that were regulated by obesity but not associated with TC, using 16 out of 91 TC array-expression datasets from Gene Expression Omnibus (GEO, https://www.ncbi.nlm.nih.gov/geo/). During this step, both the fixed-effect model and random-effects model were employed to study the effect size of the selected genes in a case *vs.* control expression comparison. The expression log fold change (LFC) was used as the effect size. Results from both models were reported and compared. The heterogeneity of the mega-analysis was analyzed to study the variance within and between different studies. In the case that the total variance Q is equal to or smaller than the expected between-study variance df, the statistic $ISq = 100\% \times (Q-df)/Q$ will be set as 0, and a fixed-effect model was selected for the mega-analysis. Otherwise, a random-effects model was selected. The Q-p represents the probability that the total variance is coming from within-study only. Significant genes from the mega-analysis were reported, which were identified with the criteria as follows: *p-value* $< 10^{-7}$ and abs (effect size (log fold change)) >1. All analysis was conducted by an individually-developed MATLAB (R2017a) mega-analysis package. We used the term 'mega-analysis' rather than 'meta-analysis' due to the fact that the log-fold changes of each gene were calculated from the original datasets.

## Literature-based pathway analysis and GSEA

For the possible risk genes identified through the expression mega-analysis described above, a literature-based pathway was constructed to identify the connection between the target genes and the TC. The analysis was performed using the 'Shortest Path' module of Pathway Studio (http://www.pathwaystudio.com). Then all the molecules within the identified networks were tested using a GSEA analysis against the Gene Ontology (GO) terms and Pathway Studio pathways. Significantly enriched pathways and corresponding statistics were reported.

## Multiple linear regression analysis

The MLR model was employed to study the possible influence of three factors on the gene expression change in TC: sample size, population region, and study date. *P-values* and 95% confidence interval (CI) were reported for each of the factors.

**Table 1  The datasets used for gene-TC relation mega-analysis.**

| Dataset GEOID | nControl | nCase | Study region | Study age |
|---|---|---|---|---|
| GSE35570 | 51 | 65 | Poland | 4 |
| GSE58545 | 18 | 27 | Poland | 4 |
| GSE58689 | 18 | 27 | Poland | 4 |
| GSE60542 | 34 | 33 | Belgium | 4 |
| GSE65144 | 13 | 12 | USA | 4 |
| GSE39156 | 16 | 48 | Belgium | 6 |
| GSE53157 | 3 | 24 | Portugal | 6 |
| GSE29265 | 20 | 29 | Belgium | 7 |
| GSE33630 | 45 | 60 | Belgium | 7 |
| GSE27155 | 4 | 95 | USA | 8 |
| GSE5364 | 58 | 270 | Singapore | 11 |
| GSE6339 | 135 | 48 | France | 12 |
| GSE9115 | 4 | 15 | USA | 12 |
| GSE3678 | 7 | 7 | USA | 13 |
| GSE6004 | 4 | 14 | USA | 13 |
| GSE3467 | 9 | 9 | USA | 14 |

# RESULTS

## Common genes for obesity and TC

As presented in the Obesity_TC database, there were 1,036 genes associated with TC and 534 influenced by obesity. A significant overlap of 176 genes was identified for both obesity and TC (Right tail Fisher's Exact test $p$-value $= 4.07e−112$), which counts for about one-third of the obesity-regulated genes (32.96%). For detailed information on these genes, please refer to Obesity_TC.

## The selected gene expression datasets

There were 16 datasets satisfied the selection criteria and were included for the mega-analysis, as shown in Table 1. According to the approach that we acquired the disease-gene relation data (by using Pathway Studio; http://www.pathwaystudio.com), about 67.04% of the obesity-genes (358 out of 534 genes) have not been reported to have an association with TC. Thus, we tested the expression changes of these 358 genes in the case of TC.

## Mega-analysis results

There were five genes (i.e., FABP4, CFD, GHR, TNFRSF11B, and LTF) passed the significance criteria ($p$-value $< 10^{-7}$ and abs (LFC)>1), which were provided in Table 2. We presented the mega-analysis results of all obesity-regulated genes in Obesity_TC →Mega_Analysis. There were four other genes (TMEM173, PLA2G7, SOD3, and AGTR1) that showed less significance ($p$-value $< 7.08e−6$) but also with a big change in terms of LFC (abs (LFC)>1).However, the discussion and analysis here were focused on the five genes that passed the significance criteria. The LFCs of the genes were estimated from the majority of the studies: 15 out of 16 studies. Notably, the Random-effects model was

**Table 2  Significant Obesity-genes from mega-analysis for TC.**

|  | Gene name | FABP4 | CFD | GHR | TNFRSF11B | LTF |
|---|---|---|---|---|---|---|
| **Mega-analysis Results** | Using random effects model | 0 | 1 | 1 | 0 | 0 |
|  | #Study | 15 | 15 | 15 | 15 | 15 |
|  | Effect size (LFC) | −1.83 | −1.79 | −1.70 | −1.50 | −1.44 |
|  | $p$-value | 6.4E−09 | 2.03E−08 | 3.61E−09 | 3.24E−09 | 4.17E−09 |
|  | nSample | 0.22 | 0.078 | 0.53 | 0.11 | 0.0012 |
| **MLR Results** | Country | 0.0019 | 1.01E−05 | 2.24E−05 | 0.033 | 0.029 |
|  | StudyAge | 0.46 | 0.94 | 0.98 | 0.37 | 0.14 |

used for CFD and GHR, and the fixed-effect model was selected for FABP4, TNFRSF11B, and LTF. MLR results showed that the age of studies and the sample sizes presented no significant influence on the effect size (LFC) of all five genes except LTF ($p$-value > 0.05), but the sample's population region (country) was a significant factor for all of them ($p$-value < 0.033, Table 2).

## Literature-based pathway analysis

To explore the functional association between the five obesity-regulated molecules and TC, we conducted a literature-based functional network analysis and presented in Fig. 1 the identified pathways. Results showed that genes GHP, TNFRSF11B, and LTF could be inhibitors of TC, through the stimulation of TC inhibitors or deactivation of TC promoters. In the case of obesity, the activity of these molecules was suppressed. On the contrary, CFD and FABP4 were suggested as two facilitators of the pathological development of TC. CFD stimulates INS, which is a promoter of TC. FABP4 inhibits three TC suppressors, including BCL2, PTEN, and PPARG. Notably, obesity activates these two molecules. The pathways revealed in Fig. 1 suggested possible mechanisms of the TC-promoting effect of obesity. For the supporting references of the relationships presented in Fig. 1, please refer to TC_Obsesity →ShortestPath.

## GSEA results

To understand the functional profile of the 14 genes involved in the pathways presented in Fig. 1, we conducted a GSEA against GO terms and Pathway Studio Pathways (http://www.pathwaystudio.com) and presented the top 10 results in Table 3. The full list of 31 pathways/GO terms enriched with $p$-value < 0.005 ($q = 0.005$ for FDR correction) has been listed in TC_Obsesity →GSEA. Notably, all the 14 genes were involved in the 31 pathways, and 12 out of 14 were included in the top 10 pathways.

Based on the GSEA results, we analyzed the connection of the five potential TC-genes and nine of their targets presented in Fig. 1, in terms of their shared pathways, as shown in Fig. 2. The number in a cell represents the number of shared pathways/GO terms by the two corresponding genes, and a number on the diagonal represents the number of pathways enriched by the specific gene. As shown in Fig. 2, most of these molecules play roles together with other molecules in multiple pathways, indicating they were functionally connected.

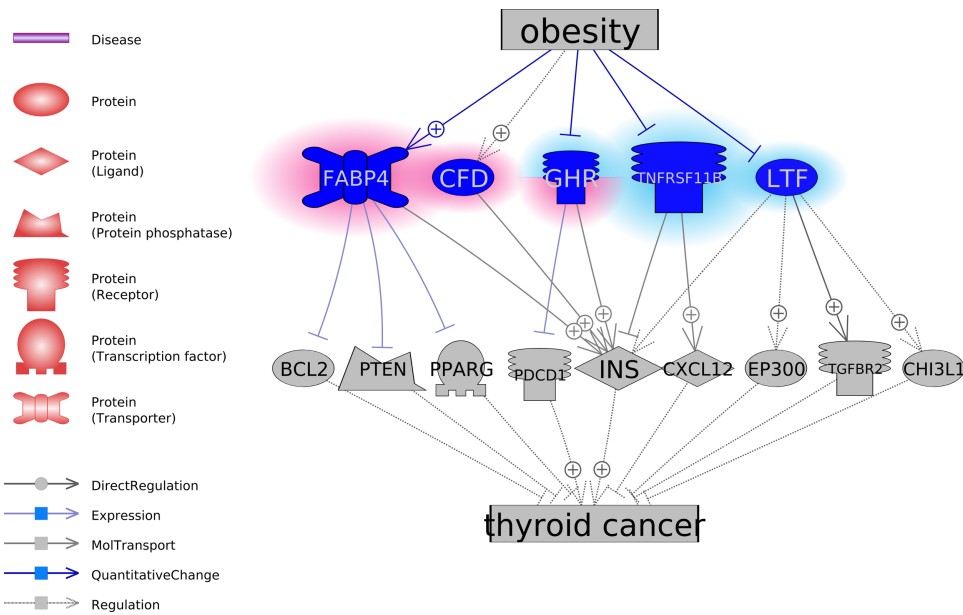

**Figure 1** The potential pathways connecting the five obesity-regulated genes and thyroid cancer.

**Table 3** The top 10 enriched pathways/GO terms by the 14 genes in regulating pathway identified in Fig. 1.

| Name | GO ID | # of Entities | Overlap | Overlapping entities | FDR corrected *p*-value |
|---|---|---|---|---|---|
| Regulation of growth | 0040008 | 887 | 8 | PPARG;BCL2;PTEN;GHR; INS;CXCL12;TGFBR2; EP300 | 0.00028 |
| Response to estrogen | 0043627 | 155 | 5 | PPARG;TGFBR2;EP300; TNFRSF11B;BCL2 | 0.00028 |
| Leukocyte activation | 0045321 | 991 | 8 | BCL2;INS;CHI3L1;CFD; CXCL12;LTF;TGFBR2; EP300 | 0.00028 |
| Positive regulation of growth | 0045927 | 354 | 6 | INS;BCL2;CXCL12; TGFBR2;EP300;GHR | 0.00028 |
| Response to nutrient levels | 0031667 | 730 | 7 | PPARG;BCL2;PTEN;GHR; TNFRSF11B;INS;TGFBR2 | 0.00052 |
| Response to extracellular stimulus | 0009991 | 761 | 7 | PPARG;BCL2;PTEN;GHR; TNFRSF11B;INS;TGFBR2 | 0.00052 |
| Regulation of developmental growth | 0048638 | 445 | 6 | BCL2;PTEN;CXCL12;GHR; TGFBR2;EP300 | 0.00052 |
| Response to glucose | 0009749 | 219 | 5 | INS;BCL2;PTEN;TGFBR2; EP300 | 0.00052 |
| Response to alcohol | 0097305 | 460 | 6 | PPARG;BCL2;PTEN;GHR; TGFBR2;EP300 | 0.00052 |
| Response to hexose | 0009746 | 227 | 5 | INS;BCL2;PTEN;TGFBR2; EP300 | 0.00052 |

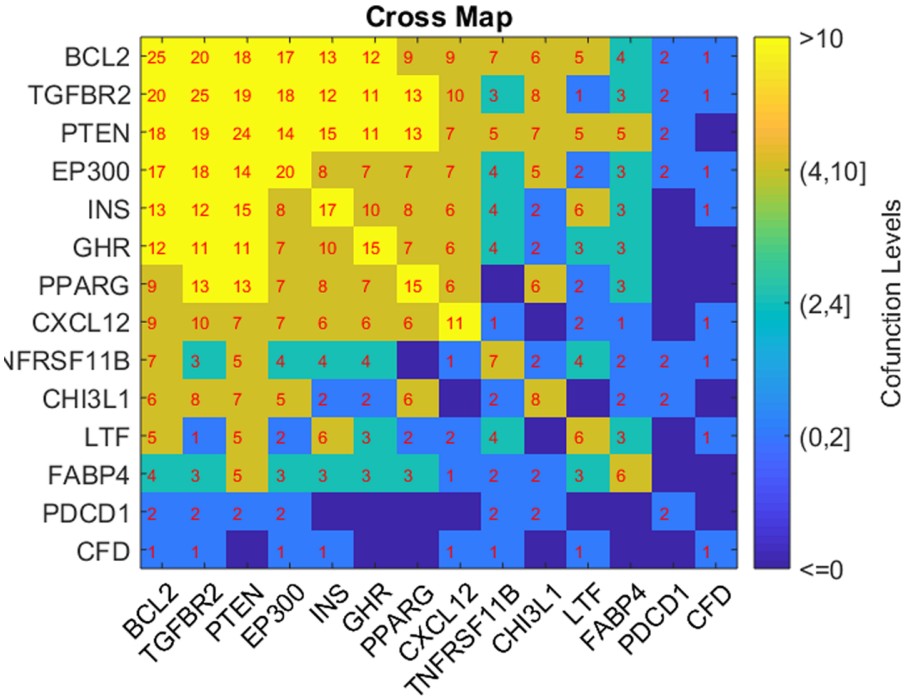

**Figure 2** **Heat map of the shared pathways by the five potential TC-genes and nine of their targets.** The number in a cell represents the number of shared pathways/GO terms by the two corresponding genes.

## DISCUSSION

In this study, we attempted to explore the mechanism underlying the TC-promoting effect of obesity at the genetic level. Towards this purpose, we first used the knowledge-based algorithms to analyze disease-gene relation data and reveal 176 obesity-regulated genes. Identified genes were also related to TC and utilized to build a common background at the genetic level for the etiology of both obesity and TC. We also uncovered 358 obesity-regulated genes that have not been implicated with TC. To test the potential connection between each of the 358 obesity-regulated genes and TC, we queried and selected the qualified TC-RNA expression datasets from GEO (https://www.ncbi.nlm.nih.gov/geo/), then we conducted a mega-analysis. Five genes were suggested as novel targets for the development of TC, including FABP4, CFD, GHR, TNFRSF11B, and LTF (see Table 2; $p\text{-value} < 10^{-7}$ and LFC $<-1.44$).

Notably, we used log fold change (LFC) instead of original expression levels for the mega-analysis, which was calculated as the expression levels of the expression level of TC patients over the mean of the expression level of healthy controls, followed by log2 transformation. We assume that, within the same dataset, patients and controls shared a similar background. Thus, by using the LFC, the influence of the background noise was minimized. In addition, we conducted a heterogeneity test for each gene, and a random-effects model was used in the case there was a significant between-study variance such that the study-specific expression variances were taken into account.
FABP4 encodes the fatty acid binding proteins that bind long-chain fatty acids and other hydrophobic ligands. The roles of FABPs include fatty acid uptake, transport, and metabolism (*Furuhashi et al., 2015*). FABP4 has been showed to induce proteasome degradation of PPAR $\gamma$ (*Nishina et al., 2017*), and decrease the expression of PTEN (*Jin et al., 2018*) and BCL2 (*Yao et al., 2015*). Reduced expression of PPAR $\gamma$, PTEN, and BCL2 family proteins have been shown to play critical roles in the pathologic development TC (*Copland et al., 2006*; *Leonardi et al., 2012*; *Gunda et al., 2017*). These pathways in Fig. 1 suggested FABP4 as a facilitator of the development of TC through the down-regulation of its inhibitors.

CFD encoded protein adipsin that stimulates the secretion of insulin (*Lo et al., 2014*), which has been suggested to play roles in the proliferation of TC cells by promoting insulin-like growth factor (*Oberman et al., 2015*). Therefore, CFD could be a direct promoter of TC. Add together the fact of increased protein levels of FABP4 and CFD in obese patients (*Cabré et al., 2012*; *Kwon et al., 2012*) could partially explain the contribution of obesity to TC.

On the contrary, pathways analysis suggested GHP, TNFRSF11B, and LTF as potential TC inhibitors (Fig. 1). It has been shown that GHP inhibits the expression of PD-1 (*Zhou et al., 2017*), which is a TC treatment target (*Bi et al., 2019*). TNFRSF11B stimulates the secretion of CXCL12 (*Benslimane-Ahmim et al., 2011*), which was suggested to contribute to TC development by regulating cancer cell migration and invasion (*Zhang et al., 2017*). Finally, the reduced TGF-beta Type-II receptor (TGFBR2) mRNA was shown to play a role in the pathogenesis of papillary TC (*Matoba et al., 1998*), while LTF interacts with TGFBR2 to activate TGF-$\beta$ signaling and initiates the formation of TbRIII:TbRII:TbRI complex (*Jang et al., 2015*). Thereby, decreased levels of GHP, TNFRSF11B, and LTF in obesity could facilitate the pathologic development of TC.

GSEA analysis showed that the five genes (FABP4, CFD, GHR, TNFRSF11B, and LTF) and nine of the downstream target genes mainly played roles in the cell growth related signaling pathways (Table 3). Notably, all these five genes regulate the insulin (INS) levels positively or negatively, while INS level was related to proliferation and carcinogenesis of TC cells (*Oberman et al., 2015*; *Malaguarnera et al., 2017*). Our results suggested that obesity may partially affect the pathologic development of TC through its influence on the INS levels.

Moreover, these genes demonstrated a robust functional connection in terms of shared common pathways (Fig. 2). The relationship between TC and the nine target genes (INS, BCL2, PTEN, PPARG, PDCD1, CXCL12, EP300, TGFBR2, and CHI3L1) were supported by previous studies (see TC_Obesity $\rightarrow$ ShortestPath), which supports the potential association between the five obesity-regulated genes (FABP4, CFD, GHR, TNFRSF11B, and LTF) and TC.

Nevertheless, this study has several limitations that can be addressed in the future work. First, the connections between the TC and the five obesity genes were suggested by mega-analysis and explored by literature-based pathways analysis. Biologic experiments are needed to test these relationships. Second, due to the lack of space, the discussion was focused on the five genes that passed the significance criteria in the mega-analysis. However,

more genes with less significance may also be worthy of inspection (e.g., TMEM173, PLA2G7, SOD3, and AGTR1). Third, further validation of the relationships between the five target genes, insulin, TC, and obesity, can be done using other tools and data sources (e.g., Hetionet v1.0; https://neo4j.het.io/browser/).

## CONCLUSIONS

Using the system biology approach, we mined a set of genes influenced by obesity to uncover five genes (FABP4, CFD, GHR, TNFRSF11B, and LTF) as previously unrecognized contributors to the development of TC. An analysis of functional network built upon these genes points towards INS as a remarkable bridging factor connecting obesity and TC.

### Funding
The authors received no funding for this work.

### Competing Interests
The authors declare there are no competing interests.

### Author Contributions
- Jiaming Chen performed the experiments, analyzed the data, prepared figures and/or tables, and approved the final draft.
- Hongbao Cao and Meng Lian conceived and designed the experiments, authored or reviewed drafts of the paper, and approved the final draft.
- Jugao Fang conceived and designed the experiments, prepared figures and/or tables, and approved the final draft.

### Data Availability
    Raw data is available as Supplemental File.

### Supplemental Information
Supplemental information for this article can be found online at http://dx.doi.org/10.7717/peerj.9302#supplemental-information.

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
