# Peer review of "Five genes influenced by obesity may contribute to the development of thyroid cancer through the regulation of insulin levels"

_PeerJ, doi:10.7717/peerj.9302_

## Round 0.1 · original submission · Major Revisions

All issues pointed by both reviewers should be addressed and the manuscript should revised accordingly.

Reviewer 1 ·

Basic reporting

no comment

Experimental design

no comment

Validity of the findings

no comment

Additional comments

This study employed large-scale literature data and meta-analysis to explore novel connections to explain the fact that obesity could increase the risk of thyroid cancer (TC). Their approach identified five genes (i.g., FABP4, CFD, GHR, TNFRSF11B, and LTF) that were significantly down-regulated in the case of TC, and these genes have been shown to be regulated by obesity but have not been implicated with TC, which makes the finding novel and thus important to the field of study. However, I do have several suggestions and concerns, as listed as follows.

It seems that the authors conducted a systematic literature data mining to extract genes that were implicated with TC and also influenced by obesity (176 genes). A total number of 1036 genes were identified to be associated with TC, and 534 are downstream targets of obesity. This is valuable information. However, how they did the data mining and what rules are applied to define an ‘association’ is not clearly described. How to qualify the data? Is there any confidence score for the identified genes?
Regarding the mega-analysis, the selection of the dataset is ok. However, I wonder why the authors use the term mega-analysis instead of meta-analysis? This is not clear to me.
Regarding the criteria for selecting significant genes from mega-analysis, p-value<10-7, and LFC>1 were used. While this criterion is reasonable, there should be more ‘less’ significant genes that should be addressed in the discussion. I see that the authors listed this as a limitation of the study. However, I suggest the author gives more statistical description of the mega-analysis results. For example, with abs(LFC)>1, how many were with p-value<0.05, how many with p-value<0.001, and so on. This will give the audience a better picture of the expression data for the mega-analysis results.
Regarding the pathway (Figure 1), I cannot find the reference information for LTF->INS in the supplementary material (Obesity_TC), while I see this one in the figure. And it seems that this relation is the only one with no polarity, please explain and provide corresponding information in the supplementary materials.
The author concluded that (in the Abstract as well as the Conclusion Sections) ‘Obesity may influence the pathologic development of TC through the regulation of the upstream regulators of TC.’ This statement, especially in the Abstract, seems redundant with other statements and suggest to remove it.

Minor points:
1. Please add references for the statement given in the lines 246 to 248: ‘the reduced TGF-beta Type-II receptor (TGFBR2) mRNA was shown to play a role in the pathogenesis of papillary TC, while LTF interacts with TGFBR2 to activate TGF-β signaling and initiates the formation of TbRIII:TbRII:TbRI complex. ‘

·

Basic reporting

The paper was clear and concise, and it fits to the basic reporting criteria. Literature was well referenced & relevant with the context. You might want to talk more about mega-analysis and why you chose this method instead of other methods in the literature. Figures are relevant but, if you can, you should increase the resolution on Figure 2 for the actual publication.
In line 159, you mentioned “About 67.04% of the obesity-genes (358 out of 534 genes) have not been reported to have an association with TC. Thus, we tested the expression changes of these 358 genes in the case of TC.” You might want to talk more in detail about how you obtained these numbers (I am assuming you used Pathway Studio.)

Experimental design

I have found no problems with your study design, but I have some concerns about background correction for the different datasets you used. Did you take any measures to nullify the effects of different studies introducing noise to your findings? You might want to talk about this issue in your discussions to make a stronger point and prove the validity of your findings.

Validity of the findings

The comments and validity of findings are in the pdf file

Additional comments

Overall, a great paper. Just needs some clarifications. I have made extra suggestions for Discussion through navigating another database i have experience working with. I wish you good luck for the future work and publications.

---

## Round 0.2 · accepted · Accept

Since all critiques were adequately addressed and the manuscript was amended accordingly, the revised version is acceptable now.

Reviewer 1 ·

Basic reporting

The authors answered all my questions that I make during last review, and make changes accordingly. I have no further questions.

Experimental design

Good in its current form.

Validity of the findings

Good in its current form.

Additional comments

The authors answered all my questions that I make during last review, and make changes accordingly. I have no further questions.